# Analysis of ^222^Rn Surface Concentrations in the Basque Country (Spain): A Case Study of Heat Waves

**DOI:** 10.3390/ijerph20032105

**Published:** 2023-01-24

**Authors:** Natalia Alegría, Miguel Ángel Hernández-Ceballos, Giorgia Cinelli, Igor Peñalva, Jose Miguel Muñoz

**Affiliations:** 1Department of Energy Engineering, University of the Basque Country, 48013 Bilbao, Spain; 2Department of Physics, University of Cordoba, 14071 Córdoba, Spain; 3Laboratory of Observations and Measurements for the Climate and the Environment, National Agency for New Technologies, Energy, and Sustainable Economic Development (ENEA), 21027, Ispra, Italy; 4Department of Industry, Basque Government, 01003 Vitoria, Spain

**Keywords:** radon, heatwave, northern Iberian Peninsula, air masses, local meteorology, climate change

## Abstract

The objective of this study was to characterize radon concentrations registered in the Radiological Surveillance Network of the Basque country in relation to local meteorological parameters, and to determine its behaviour under heatwave events. For this purpose, radon measurements and meteorological parameters from June 2012 to June 2015 were analysed at two sites, Bilbao and Vitoria (northern Spain), in a region characterized by complex orography, causing large temporal and spatial variability in meteorological conditions. Yearly, seasonal, and diurnal cycle differences and similarities were investigated at both sites. The temporal evolution of radon concentration was analysed at both sites during the two heatwave periods officially identified by the State Meteorological Agency (8–11 August 2012 and 17–23 August 2012). The analysis revealed two different patterns of radon concentrations, in terms of both time and intensity, under this synoptic pattern, making it also possible to identify regional transport channels of radon concentrations between the two sites. This set of results evidences the adequate position of both stations to represent the spatial and temporal evolution of radiological variables continuously in this region.

## 1. Introduction

Networks for radiological monitoring are designed to monitor the radiological quality of the environment and detect, possible incidents in their early phases, thus identifying nuclear and radioactive releases from nuclear and radioactive sites, as well as delineating the evolution of the radioactivity plume [1]. These networks, are therefore vital to evaluating the background levels of environmental radiation [2], determining the causes of radiation levels in the environment [3], and guaranteeing compliance with the legal and regulatory requisites imposed [1].

The naturally radioactive noble gas ^222^Rn (hereafter referred to as radon) is one of the radiological variables continuously monitored all around the world [4] due to the fact that, in combination with its progeny, it accounts for about 50% of the dosage due to sources of natural radioactivity [5]. It is a naturally occurring gas produced by radioactive decay of the ^238^U progeny in rocks and soils [6]. Once produced in soils/rocks, radon can reach the atmosphere or be accumulated in buildings/caves/etc., where it can represent a health risk. Indeed, radon exposure is, after tobacco smoke, one of the major causes of lung cancer [7]. Levels of radon must be constantly monitored to prevent excessive exposure and protect the population, as required by legislation [8]. However, there are also some positive aspects, because radon measurements are relevant in different scientific disciplines [9,10]. For example, atmospheric radon can be used as a tracer to improve atmospheric transport models [11,12], and can be used together with radon flux to indirectly estimate greenhouse gas (GHG) fluxes using the Radon Tracer Method (RTM) [13,14]. Due to the importance of this kind of measurement, research is ongoing with the EMPIR project 19ENV01 traceRadon (http://traceradon-empir.eu/), which started in 2020, with the aim of improving traceable low-level radon activity concentration and radon flux measurements [15]. 

In Spain, the continuous monitoring of the concentration of radon progeny is the responsibility of the Spanish Nuclear Safety Council (Consejo de Seguridad Nuclear, CSN), in collaboration with its equivalent organs in the autonomous communities. These measurements are performed by the automatic stations network (REA), which consists of 25 monitoring stations all over the Spanish region, which perform measurements in real time [16]. 

In this context, and from the beginning of the 2000s, the Basque Government (GV) has developed an environmental radiological monitoring programme through the progressive deployed of radiological stations. At these stations, in addition to measuring radon concentrations, meteorological variables are also measured. Previous studies have analysed outdoor radon concentration measurements in Bilbao [17], as well as indoor radon concentrations in Pozalagua cave [17,18]. However, no studies have yet addressed the spatial variability of radon measurements in this region. For this reason, the first objective of this paper is to characterize radon concentrations in the Basque Country, taking as a reference two monitoring sites, Bilbao and Vitoria-Gasteiz, which are only approximately 70 km distant from one another, but present clear meteorological differences due to their different proximities to the coast (Bilbao is about 15 km, and Vitoria is about 65 km), and the physical separation between them (i.e., the Gorbea mountain chain). 

This analysis also makes it possible to investigate the behaviour of ^222^Rn concentrations under different meteorological scenarios. In this sense, and in the context of global warming, heatwaves are an increasingly important weather phenomenon due to their significant contribution to air quality, heat-related morbidity and mortality [19,20]. Scientific efforts to understand the characteristics and influence of this phenomenon have been increasing in recent years [21,22], with climatic projections predicting extreme heatwaves that are more intense, prolonged and frequent in Europe in the 21st century, with a higher impact on the Iberian Peninsula and Mediterranean regions [23]. To the best of the authors’ knowledge, no study has been conducted investigating the behaviour of ^222^Rn concentrations under heatwave conditions. For this reason, the second objective of the present paper was to analyse the temporal variability of ^222^Rn concentrations under heatwave events registered in the Basque Country in order to evaluate their behaviour during these extreme meteorological events. 

Our present analysis therefore aims at addressing the following research issues: Analyse and compare the three-year times series and investigate the seasonal and daily cycles of radon concentration in Bilbao and VitoriaInvestigate the dependence of hourly concentration at the surface on wind direction and speed.Perform an analysis of radon time series under heatwave events at both sampling sites and provide explanations.

The structure of the paper is as follows: The methodology and data are summarised in Section 2 and Section 3, while Section 4 presents and discusses the results obtained. Section 5 summarises the most relevant conclusions following the objectives presented.

## 2. Study Area

The Basque Country, in the northern part of the Iberian Peninsula, is mainly characterized by rugged terrain is shown in Figure 1. This region is bounded by the Bay of Biscay to the north and by the Ebro River to the west and south side. In the middle, there are the mountains along the east–west axis, while the Pyrenees Mountains separate the region from France to the northeast. These geographical features mean that, in terms of climate, the Basque Country is not a homogeneous area, and therefore, Bilbao (43, 25 N; −2.92 W; 35 m above sea level) and Vitoria (42.86 N; −2.67 W; 625 m above sea level), despite only being separate by 70 km, belong to different climatic zones. While Bilbao, to the north, belongs to the Atlantic seaboard, with heavy rainfalls and mild temperature, Vitoria is located in an intermediate area with continental climate characteristics. 

According to MARNA, which is the map of natural gamma radiation in Spain [24], both areas can be classified as possessing low natural background radioactivity based on the values of U contents in soil, gamma dose rates, and radon fluxes. The Bilbao area and the Vitoria area are indicated to possess similar uranium values, between 1.6 and 2.4 mg/kg, in the European Map of Uranium in soil (https://remon.jrc.ec.europa.eu/About/Atlas-of-Natural-Radiation/Digital-Atlas/Uranium-in-soil/Uranium-concentration-in-soil- accessed on 1 October 2022). 

## 3. Materials

In this section, we provide a description of the data and map used in the analysis. The HYSPLIT model, which was used to calculate the set of air mass trajectories, is also described. 

### 3.1. Data

After the Chernobyl incident, and following the indication of the CSN, the Basque Government deployed a Berthold BAI 9100D in Vitoria for the purposes of controlling and monitoring the contamination risks from the atmospheric dispersion of radionuclides possible released from the Santa María de Garoña NPP, which is located 70 km away. After this, a second measuring site was located in Bilbao, close to the sea. The purpose was, in combination with CSN, to make it possible to manage the nuclear and radioactive emergency response, perform severe accident management, as well as the management of security and knowledge. 

Radon is measured by the BAI9100 using the ABPD-Compensation and Measurement principle. The radiological station located in Bilbao and Vitoria is a combined Particulate and Iodine Monitoring system Type 9850-6 (Moving Filter Monitor BAI 9100 D, IOD-131 monitor BAI9103-2, and Gamma Dose Rate Detector LB6360, Berthold trademark) positioned on the roof of a building 10 m above ground level. Briefly, radon is estimated on the basis of its progeny. Since equilibrium factor measurements were not available, only the output of the instrument was considered, while noting that, when referring to ^222^Rn concentration, this refers to ^222^Rn as estimated on the basis of its progeny. The reader is referred to [25,26] to obtain more information about how the radon time series are obtained. In the present study, radon measurements, registered continuously at 10 min intervals from June 2012 to June 2015, were used. This period was selected based on the high number of radon measurements that were taken at the same time in both Bilbao and Vitoria. Hourly values were also calculated in the present analysis while applying the sufficiency criterion (at least 75% of the 10 min and hourly values). 

The time series of the following meteorological parameters were also used: wind speed (m/s) and direction (°), and temperature (°C). Meteorological stations at both sampling sites used CR1000 as a datalogger, and the rest of the sensors were of the Campbell trademark [27].

### 3.2. Air Mass Trajectories and Weather Charts

Backward trajectories and the geopotential height of 500 hPa were used to characterize heatwave events. These upper-level charts were taken from the Wetterzentrale webpage (https://www.wetterzentrale.de/ accessed on 31 October 2022), and illustrate the dynamics of our atmosphere, showing the predominant tropospheric waves (trough or ridge), which virtually control the weather at the surface (dry, warm/wet, cold). The Hybrid Single Particle Lagrangian Integrated Trajectory (HYSPLIT) model [28], using GDAS meteorological files (1 degree), was used to calculate a set of twelve 3-D backward trajectories of 96 h per day at initial heights of 1500 and 3000 m above ground level. 

## 4. Results and Discussion

### 4.1. ^222^Rn Concentrations in Basque Country (June 2012–June 2015)

When working with data, ideally, they should be acquired continuously and without gaps; however, this is almost impossible in this case due to power failures or other technical problems. In this sense, and in order to perform a comparison of ^222^Rn concentrations between the two sampling stations, we defined reference “pairs of days”. First, we identified those days in Bilbao and Vitoria with a sufficient number of hourly ^222^Rn data, i.e., those days with more than 18 hourly data points out of 24 (75%). Those days with a percentage below this threshold were discarded for the present analysis. Then, we defined “pairs of days” as the set of non-value pairs between Bilbao and Vitoria, so that pairs of days in which either Bilbao or Vitoria were null were not included in the analysis. This method implies the selection of 660 “pairs of days” during the period June 2012 to June 2015 to be able to perform a comparison of ^222^Rn concentrations between Bilbao and Vitoria. The seasonal distribution of these days, 171 in winter, 143 in spring, 149 in summer and 197 in autumn, guarantees the largest statistical sample.

Figure 2 shows the ^222^Rn time series (1 h) and Figure 3 displays the total and seasonal box plots for hourly concentrations during the period June 2012–June 2015 for Bilbao and Vitoria. During this period, ^222^Rn concentration averaged 10.7 ± 0.1 Bq m^−3^ in Bilbao and 10.1 ± 0.1 Bq m^−3^ in Vitoria, with the uncertainty being provided as the mean standard deviation *Sx*/*N*^1/2^ (being *Sx* is the standard deviation and *N* is the number of sampling sites). These averages are in agreement with the typical outdoor radon concentration, which is between 1 and 100 Bq m^−3^, and with the estimated annual average of around 10 Bq m^−3^ [4]. The correlation between ^222^Rn concentrations in both sites is positive (*r* = 0.70, at 0.05 significance level), which clearly indicates a similar evolution of activity concentrations. However, Figure 3 shows large differences in concentrations. Hourly concentrations were higher in Bilbao in 52% of the time, and, on average, these differences in concentrations were higher in spring (8.4 Bq m^−3^ and 7.7 Bq m^−3^), summer (11.0 Bq m^−3^ and 9.0 Bq m^−3^) and autumn (13.3 Bq m^−3^ and 12.1 Bq m^−3^), while only in winter were they higher in Vitoria (9.9 Bq m^−3^ and 9.5 Bq m^−3^) (black cross in Figure 4). The largest differences were registered in summer (2 Bq m^−3^), which can be attribtued with the development of mesoscale circulations in the coastal area, which can increase surface concentrations. Previous studies have pointed out the link between the occurrence of sea–land breezes and the variability in ^222^Rn evolution and its occurrence in high concentrations [29,30]. Specifically, in this region, in 2020 and 2021, in [31,32,33], a positive link was revealed between this mesoscale phenomenon and high beta, alpha and ^7^Be activity concentrations. Conversely, in winter, when the differences were at a minimum, the whole region was more frequently under the influence of synoptic winds, favouring cleaning processes, with Bilbao, which is located in the coastal area, also being more affected by maritime winds, thus resulting in a decrease in ^222^Rn activity concentrations. 

The distribution of ^222^Rn concentrations during the whole sampling period was wider in Bilbao than in Vitoria (Figure 3, left), and the largest differences between the two were found for the highest concentrations, i.e., the comparison shows how the percentiles above the 50th percentile in Bilbao were higher than those from Vitoria. This behaviour was also observed in spring and summer (Figure 4, right), but not in winter. Conversely, in autumn, the range of ^222^Rn concentrations was wider in Bilbao, but the highest ^222^Rn concentrations were reached in Vitoria. These results highlight the different seasonal patterns observed between Bilbao and Vitoria. Meteorological seasons were defined as follows: winter, January to March; spring, April to June; summer, July to September; and autumn, October to December. While in Vitoria, the widest distributions and highest values were registered in winter and autumn, in Bilbao, these were achieved in summer and autumn, which can clearly be attributed to the different locations of the cities, as well as the differences in the meteorological conditions between the two areas of the Basque country. This seasonal evolution, for instance, is similar to that obtained in the southern area of the Iberian Peninsula [12,29] in which autumn was the season with the highest overall concentrations.

Figure 4 shows the corresponding total and seasonal daily average ^222^Rn evolution for these two sites on the basis of the 660 “pairs of days” identified during the sampling period. Similar trends in diurnal variations, with differences in magnitude, were observed. As expected, and on average (black line), ^222^Rn follows the usual daily evolution cycle [34]. At both sites, the highest values were observed during the night (between 7:00 and 8:00 UTC), coinciding with low atmospheric boundary layer (ABL) height and stable stratified conditions limiting the transport and movement of air masses, and with low wind speed, favouring the accumulation of ^222^Rn. After sunrise, the ABL becomes unstable, and therefore gains height, diluting ^222^Rn in a larger volume of air and, therefore, causing a decrease in ^222^Rn activity concentrations. The minimum occurs between 17:00 and 18:00 UTC. 

A similar pattern could also be observed on a seasonal basis, but, on top of the differences in the intensity of the maximum activity concentrations, a clear difference could be seen in terms of the amplitude of the radon daily cycle and the time range within which the maximum concentrations occur in Bilbao and Vitoria. First, the amplitude (difference between the maximum and minimum daily concentrations) is always higher in Bilbao than in Vitoria. The highest amplitude is reached in summer (14.6 Bq m^−3^ in Bilbao and 9.6 Bq m^−3^ in Vitoria), while the minimum is in winter (6.7 Bq m^−3^ in Bilbao and 6.1 Bq m^−3^ in Vitoria). In addition, secondly, the largest time range in which the maximum was registered was in Vitoria (between 07:00 and 10:00 UTC), while in Bilbao, the time range was between 7:00 and 8:00 UTC. Conversely, the minimum concentrations were registered between 17:00 UTC and 19:00 UTC at both sampling sites. These differences in amplitude and time reinforce the influence of different meteorological conditions, affecting the temporal variability of ^222^Rn activity concentrations in both sites.

Several atmospheric processes can contribute to the variations in the atmospheric ^222^Rn concentrations. In this sense, ^222^Rn can be horizontally transported by advection to a measurement site from far away, because of its half-life of 3.8 days. The wind roses depicting hourly surface wind directions and ^222^Rn concentrations for Bilbao and Vitoria throughout the whole sampling period are presented in Figure 5 to characterize the link between the two variables. Wind roses are mostly used in fields such as environmental impact assessment [35,36]. Here, data were grouped into 16 wind-direction sectors and six radon concentration intervals. To analyse these results, it is necessary to focus attention on the main orographic characteristics of the study area (Figure 1). These wind roses are in line with the orographic barriers identified in Figure 1. Two main wind directions are highlighted at each site. While in Bilbao, the maximum concentrations were registered in winds from the south/southwest (continental areas), in Vitoria, winds from the southwest/west were those responsible of the highest concentrations of ^222^Rn activity. In the context of this behaviour, it is interesting to point out that the arrival of northwesten winds over Bilbao, despite their maritime origin, is associated with relatively high ^222^Rn values. Considering the main wind directions and the associated ^222^Rn concentrations, it is possible to suggest the possible transport of ^222^Rn, under specific meteorological scenarios, from Vitoria to Bilbao with the arrival of southern winds. This meteorological scenario clearly coincides with the occurrence of heatwaves in this area [37], so the analysis of radon concentrations under this scenario can improve our understanding of its temporal variability. 

### 4.2. August 2012 Heat Waves: Synoptic Scenario

The State Meteorological Agency (AEMET) [38] identifies 18 heatwave events (102 days in total) in the Basque country during the period 1975–2021, with all of them having been registered in the province of Alava, in which Vitoria is the main city. On average, the duration of these events was 5.6 days, and 55% of them were registered in August. Among this set of events, there were two in August 2012, i.e., 8th–11th August 2012 and 17th–23rd August 2012, that constitute the basis of the present analysis. 

During August 2012, one surge of excessively warm air affected much of southern and western Europe, from Spain to Ukraine. In Vitoria, and in both periods, both maximum and minimum temperatures increased, reaching values between 35 °C and 40 °C (Figure 6), whereby it is also necessary to mention that the duration of this type of extreme event in this region was 11 days. The synoptic configuration during both periods was characterized by the presence of a high-pressure system over Europe, and a low system to the west of the Iberian Peninsula and the British Islands (Figure 7). This configuration coincides with the omega blocking configuration, which is the frequent synoptic pattern associated with heat waves in Europe, and is characterized by a low–high–low pattern along the west–east axis. This configuration favours the presence of high-pressure systems over this area over a long time period, and also favours the displacement to the Iberian Peninsula of high-pressure systems from northern Africa. This latitudinal displacement favours the transport of tropical, warm, and dry air from the Sahara Desert in the upper atmospheric levels over Spain. Figure 7 displays the set of air mass trajectories at 1500 m and 3000 m during both periods over Vitoria. A similar set of backward trajectories were obtained for Bilbao at these heights, which are the typical heights at which Saharan air masses are observed over the Iberian Peninsula. The results obtained for both periods showed a wide predominance of southerly advection, with origin in the north of Africa, thus confirming the Saharan origin of both heat waves, which is in line with the typical origin of heat waves in the Iberian Peninsula. 

### 4.3. ^222^Rn Evolution and Analysis under Heatwaves, 2012

In order to be able to perform a detailed analysis of the radon concentrations during heatwave events, the 10 min evolution of ^222^Rn concentration at Bilbao and Vitoria is shown in Figure 7. It can be seen from the figure that the maximum concentrations of radon during the month at both sampling sites were reached during heatwaves (red boxes), thereby indicating a positive influence of this synoptic scenario on radon concentrations in this area. However, and when considering the concentrations at both sampling sites throughout the whole sampling period (June 2012–June 2015) (Figure 3), it is necessary to not that the two heatwave events were not associated with the highest radon concentrations in the area. Despite this fact, two completely different evolutions of radon concentrations were observed between the two heatwave events. During the first event, the concentrations in Bilbao presented a continuous increase and reached their maximum values in the middle of the event (on the second and third days), while in Vitoria, the radon concentrations only drastically increased on the third day, decreasing on the last day. Conversely, during the second event, the radon concentrations at both sampling sites presented a similar evolution, reaching their highest values at the beginning of the event (on the first day) and progressively decreasing during the following days. 

To analyse the influence of local meteorological conditions on atmospheric ^222^Rn concentrations, we focused on changes in surface winds. Precipitation is not included in this analysis due to it not having been registered for both periods, as well as the fact that the atmospheric boundary layer height follows the expected daily cycle behaviour (not shown). Figure 8 and Figure 9 display the evolution of radon concentrations and surface wind direction and speed during both periods in Bilbao and Vitoria, while temperature changes are presented in Figure 7. 

During the first heatwave event, the synoptic conditions favoured the progressive development of well-established sea–land breeze circulations in the coastal area in the first three days, and blowing the winds from the south during the night–early morning and from the northwest during the day. Wind speed also displayed a daily cycle, with maximum intensities of less than 3 m/s. Under this scenario, and as expected under this weak forcing condition, surface ^222^Rn concentrations registered the characteristic diurnal cycle and reached the highest values during the second and third day, which was favoured by the accumulation process. This can be associated with the role of so-called “reservoir layers”, following the description given in [39], and the fact that sea–land breezes favour the continuous on-shore recirculation of radon on a daily basis. This mesoscale pattern helps to explain the maximum concentrations, but also the high background levels occurring during this event. The daily cycles of both surface winds and ^222^Rn concentration are broken at the end of this period due to the arrival of northerly winds (northwest/northeast) with higher intensity than those in the previous days, thus favouring the dispersion of ^222^Rn concentration. Conversely, in Vitoria, the increase in ^222^Rn concentrations was delayed, and was observed on the third and fourth days. These high daily concentrations were associated with the arrival of southwesterly winds with low intensity. At the end of this heatwave period, northerly winds were registered over Vitoria, with a small delay in the starting time compared to those observed in Bilbao. These results in Bilbao and Vitoria indicate a decoupling of local wind dynamics between the two sites during the first three days, with the development of sea–land breezes in Bilbao standing in contrast to the high occurrence of southern winds over Vitoria. In this sense, changes in temperature (Figure 7) help to explain this different behaviour, whereby Vitoria registered a progressive increase in temperature associated with the arrival of continental flows, while the arrival of maritime winds during the day broke this trend in Bilbao. 

During the second heatwave event, the surface winds presented a different behaviour. In this sense, the wind data showed the frequent arrival of maritime winds from the northwest/north over Bilbao, with small periods under the influence of continental winds from the southwest. This arrival of northerly winds was also registered in Vitoria, in combination with the arrival of continental flows. At both sampling sites, the arrival of maritime winds caused a decrease in radon concentration, and only the small influence of continental winds helped to increase ^222^Rn concentration. At the end of this period, and at both sites in parallel, northerly winds swept the region, resulting in a cleaning effect in the surface atmospheric layers, and radon concentrations sank to below 5 Bq m^−3^. In this sense, these results clearly indicate a similar behaviour in surface winds at the two sampling sites, with the influence of synoptic maritime winds, helping to progressively decrease radon concentrations. This influence of maritime winds can be clearly observed in the temperature evolution (Figure 8), i.e., the highest value was registered on the first day at both sites, and after this day, temperatures decreased until the end of the period. 

## 5. Conclusions

The radon concentrations collected between June 2012 and June 2015 in the Basque country region (North of Spain) were studied. Similar averages were found for Bilbao (10.73 ± 0.08 Bq m^−3^) and Vitoria (10.1 ± 0.08 Bq m^−3^), although different seasonal patterns were registered, i.e., in Bilbao, the highest values were measured in summer and autumn, while in Vitoria, they occurred in winter and autumn. These seasonal differences were observed in the analysis of daily cycles, which showed a similar trend in diurnal variations with differences in magnitudes. The role of wind direction in controlling the variability of radon concentration over time was also investigated. Whereas in Bilbao, which is largely influenced by maritime winds, the maximum concentrations were registered under flows from the south/southwest, in Vitoria, winds from the southwest/west were those responsible of the highest concentrations of ^222^Rn activity. Finally, two specific time periods corresponding to heatwaves were used to investigate and changes and variability in radon concentration. Neither event was associated with the highest radon concentrations, in either Bilbao or Vitoria, but radon presents different temporal evolutions, and hence, it is not possible to identify a general pattern of radon concentration under the occurrence of this synoptic scenario. The present analysis demonstrated the key influence of surface winds, and specifically, the development of sea–land breeze patterns in the coastal area on the temporal evolution of radon concentrations. 

## Figures and Tables

**Figure 1 ijerph-20-02105-f001:**
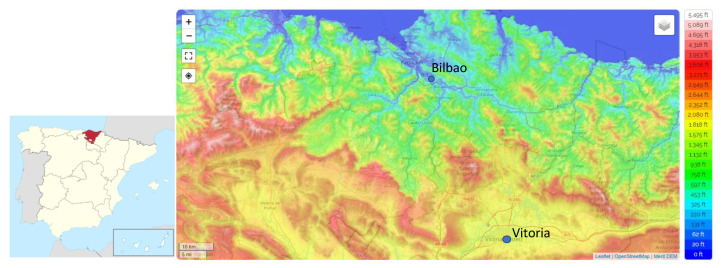
Location of the sampling site in the Basque country region (northern area of the Iberian Peninsula). Source: https://es-es.topographic-map.com/map-n3wtp/Península-ibérica/ (accessed on 14 October 2022).

**Figure 2 ijerph-20-02105-f002:**
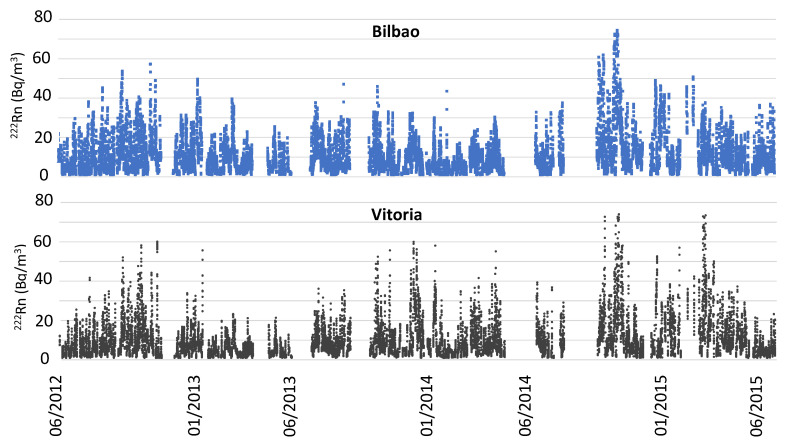
Time series of ^222^Rn hourly concentrations in Bilbao and Vitoria during the period from June 2012 to June 2015. Only hourly values of “pairs of days”, i.e., daily concentrations must both be above zero Bq m^−3^.

**Figure 3 ijerph-20-02105-f003:**
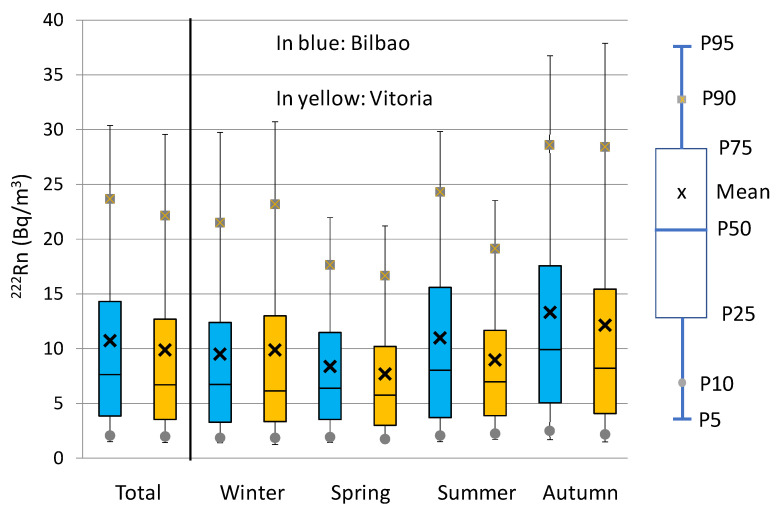
Box plots for hourly ^222^Rn concentrations during the period June 2012–June 2015.

**Figure 4 ijerph-20-02105-f004:**
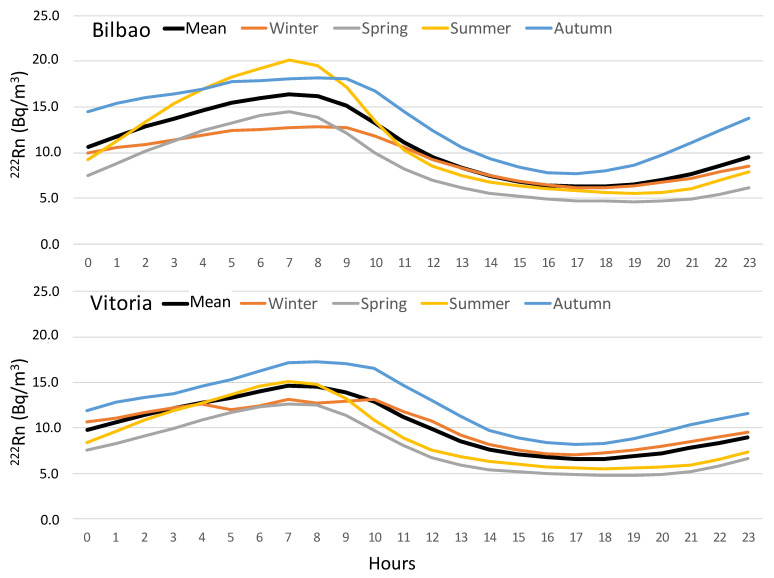
Total and seasonal daily average ^222^Rn evolution for Bilbao and Vitoria during the sampling period (June 2012–June 2015).

**Figure 5 ijerph-20-02105-f005:**
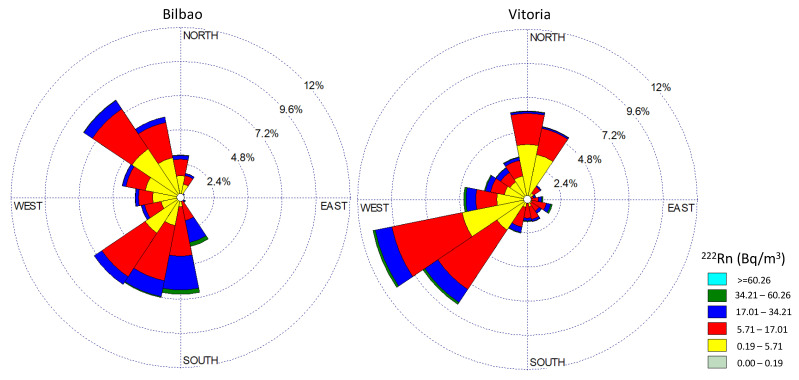
Wind roses of ^222^Rn concentration (hourly values) during the sampling period at Bilbao and Vitoria.

**Figure 6 ijerph-20-02105-f006:**
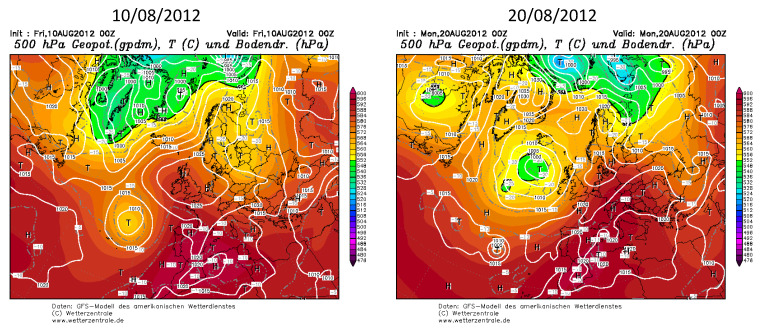
Atmospheric pressure map at 500 hPa and ground level (10/08/2012 and 20/08/2012), and the set of backward trajectories at 1500 m and 3000 m over Vitoria during both heat wave events.

**Figure 7 ijerph-20-02105-f007:**
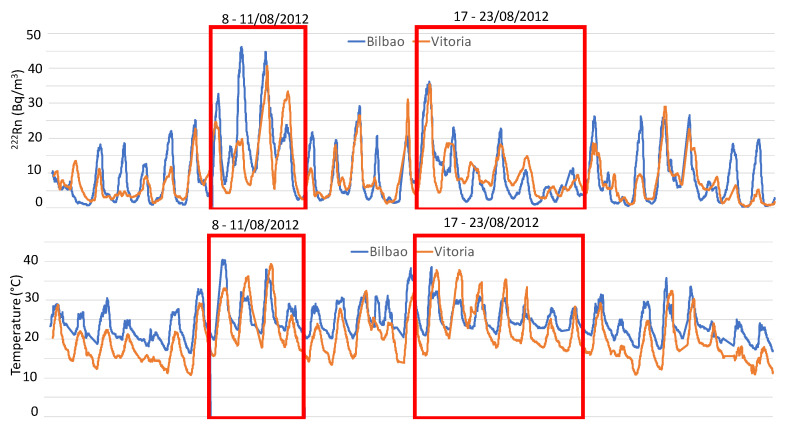
Ten-minute radon activity concentrations during August 2012 in Bilbao and Vitoria.

**Figure 8 ijerph-20-02105-f008:**
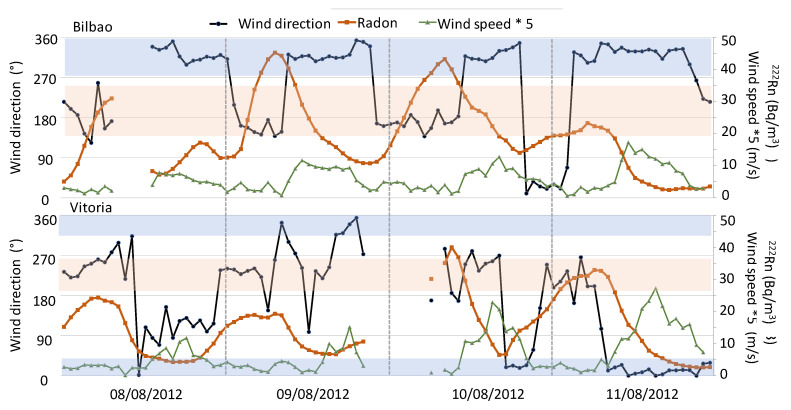
Hourly variation of wind direction and speed, and ^222^Rn activity concentration between 7th and 11th August 2012 in Bilbao and Vitoria. Note differences in scales. Blue area represents wind direction of maritime flow, while brown area represents continental flows.

**Figure 9 ijerph-20-02105-f009:**
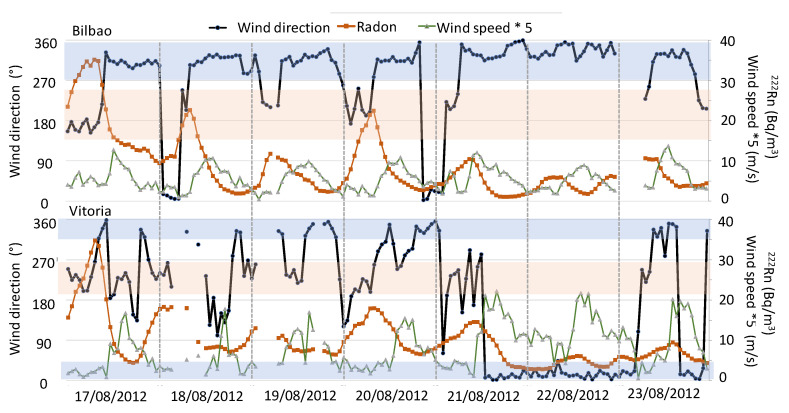
Daily variation in wind direction and speed, and ^222^Rn activity concentration between 17th and 23rd August 2012 at Bilbao and Vitoria. Note differences in scales. Blue area represents wind direction of maritime flow, while brown area represents continental flows.

## Data Availability

The data presented in this study are available on request from the corresponding author.

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
