# Peer review of "Analysis of ^222^Rn Surface Concentrations in the Basque Country (Spain): A Case Study of Heat Waves"

_ijerph, 2023, doi:10.3390/ijerph20032105_

Round 1

Reviewer 1 Report

Although the authors has presented the study well, there are some serious issues with this manuscript. 
1. The introduction section doesn’t put the study in the context of other studies like what has been done and what this study want to achieve. Specially there is no research questions answered. 
2. The materials and methods section is very limited and explains a few concepts however no details methodology on what this study want to achieve is presented which should be there 

3. The results and discussion section is presented well in detail however the discussion should present this study in the context of similar study as to what this research is improving. 
4. The references in this study are very limited which again points to limitations in consulting literature and putting the study in proper context. This needs to be addressed. 
5. For now I’ll give a major revision for this manuscript and hope the authors will improve their manuscript in later versions. 

Author Response

Reviewer 1

Although the authors has presented the study well, there are some serious issues with this manuscript. 

  1. The introduction section doesn’t put the study in the context of other studies like what has been done and what this study want to achieve. Specially there is no research questions answered. 

According with your comment, we have modified the introduction to indicate what has been done in the region, according to radon studies. We have also modified the information regarding heatwave events and radon, in order to make clear that this is the first study in which this link is analysed:

“Previous studies have analysed radon outdoor concentrations measured in Bilbao [13], as well as indoor radon in Pozalagua cave [14, 13]. However, no studies have yet addressed the spatial variability of radon measurements in this region. For this reason, the first objective of this paper is to characterize radon concentrations in the Basque Country, taking as reference two monitoring sites, Bilbao and Vitoria-Gasteiz, which are only distance 70 km but present clear meteorological differences due to its proximity to the coast (Bilbao is at about 15 km, and Vitoria at about 65 km) and the physical separation between them (Gorbea mountain chain).”

“This analysis also allows the possibility to investigate the behaviour of 222Rn concentrations under different meteorological scenarios. In this sense, and in the context of the global warming, heatwave is an increasingly important weather phenomenon due to the significant contribution to air quality, heat-related morbidity and mortality [15, 16]. The scientific effort to know the characteristics and influence of this phenomenon has been increased [17,18], i.e.,  climatic projections predict more intense, prolonged and frequent extreme heatwaves in Europe in the 21st century, with a higher impact on the Iberian Peninsula and Mediterranean regions [19]. To the best of the author's knowledge, no study has been conducted to investigate the behaviour of 222Rn concentrations under heatwaves. For this reason, the second objective of the present paper paper is to analysis the temporal variability of 222Rn concentrations under heatwave events registered in the Basque Country to evaluate potential health hazards due to radon inhalation under this extreme meteorological event.”

We have also included the following research questions to be addressed in the present study.

“Our present analysis therefore aims at addressing the following research issues:

-Analysis and comparison of the three-year time series and investigation of the seasonal and daily cycles of radon concentrations in Bilbao and Vitoria

-Dependence of hourly concentration at the surface on wind direction and speed.

-Analysis of radon time series under heatwave events in both sampling sites and provision of explanations.”

  1. The materials and methods section is very limited and explains a few concepts however no details methodology on what this study want to achieve is presented which should be there 

We agree with your comment. We have modified/included the following information in the present version of this paper:

  • We have added modified the introduction, indicating clearly which are the two objectives to be achieved
    • For this reason, the first objective of this paper is to characterize radon concentrations in the Basque Country, taking as reference two monitoring sites, Bilbao and Vitoria-Gasteiz, which are only distance 70 km but present clear meteorological differences due to its proximity to the coast (Bilbao is at about 15 km, and Vitoria at about 65 km) and the physical separation between them (Gorbea mountain chain).
    • For this reason, the second objective of the present paper paper is to analysis the temporal variability of 222Rn concentrations under heatwave events registered in the Basque Country to evaluate potential health hazards due to radon inhalation under this extreme meteorological event.

  • We have modified the information presented in the materials and methods section:
    • We have removed Figure 2
    • We have added the following information “Radon is measured in the BAI9100 by ABPD-Compensation and Measurement principle. The radiological station located in Bilbao and Vitoria is a combined Particulate and Iodine Monitoring system Type 9850-6 (Moving Filter Monitor BAI 9100 D, IOD-131 monitor BAI9103-2 and Gamma Dose Rate Detector LB6360, Berthold trademark positioned on the roof of a building at 10 m above ground level. Briefly, radon is estimated based on its progeny. Since equilibrium factor measurements were not available, it was considered simply the output of the instrument being aware that referring to 222Rn concentration it refers to 222Rn estimated through its progeny. The reader is referred to [20, 21] to obtain more information about how the radon time series are obtained. In the present study, radon measurements continuously registered at 10-min interval from June 2012 to June 2015 are used. The selection of this period is based on the large percentage of radon measurements taken at the same time in Bilbao and Vitoria. Hourly values are also calculated for the present analysis applying the sufficiency criterion (at least 75% of the 10-min and hourly values).”
  1. The results and discussion section is presented well in detail however the discussion should present this study in the context of similar study as to what this research is improving. 

We agree with your comment.

  • We have included in the introduction that this is the first study in which the link between heatwave events and radon is studied (see previous comments).
  • We have indicated in the introduction the research questions (see previous comments),
  • We have included the following information
    • These averages agree with the typical outdoor radon concentration, between 1 and 100 Bq m-3, and with the estimated annual average of around 10 Bq m-3 [4].
    • “Previous studies carried out have pointed out the link between the occurrence of sea-land breezes and the variability in 222Rn evolution and in the occurrence of high concentrations (Arrillaga et al., 2018; Gutiérrez-Álvarez et al., 2021). Specifically, in this region, Alegría et al., 2020, 2021 and Hernández-Ceballos et al., 2020 have revealed the positive link between this mesoscale phenomenon and high beta, alpha and 7Be activity concentrations.”

  1. The references in this study are very limited which again points to limitations in consulting literature and putting the study in proper context. This needs to be addressed. 

We have increased the number of references in the new version of this analysis.

Reviewer 2 Report

General thoughts

The reviewed article requires corrections and ordering

1. The title needs to be edited. Placing a date in the title is hardly justified, considering that the authors analysed the period from 2012 to 2015

2. Enrich your keywords

3. The introduction must be extensive, it is not clear from the content why this analysis is important, why the authors study these two areas and not others?

4. Methodology does not describe much of the actual methodology

5. The description of the study areas is very cursory, there is no geology of the region, geomorphology,

6. Applications must be supplemented with a comparison with the work of other researchers

7. Literature is also too modest, no confrontation with other researchers

Other comments have been placed in the comments in the attached file.

Author Response

Reviewer 3

General thoughts

The reviewed article requires corrections and ordering

  1. The title needs to be edited. Placing a date in the title is hardly justified, considering that the authors analysed the period from 2012 to 2015 DONE

We really thank your suggestion. We have changed the title of the article “Analysis of 222Rn surface concentrations in the Basque country (Spain) during heatwave events”

  1. Enrich your keywords DONE

We have completed the set of keywords in the present version of the paper ” Keywords: radon; heatwave; northern Iberian Peninsula; air masses; local meteorology; climate change.”

  1. The introduction must be extensive, it is not clear from the content why this analysis is important, why the authors study these two areas and not others?

We agree with your comment. We have modified introduction in order to point out the importance of the present study:

“Previous studies have analysed radon outdoor concentrations measured in Bilbao [13], as well as indoor radon in Pozalagua cave [14, 13]. However, no studies have yet addressed the spatial variability of radon measurements in this region. For this reason, the first objective of this paper is to characterize radon concentrations in the Basque Country, taking as reference two monitoring sites, Bilbao and Vitoria-Gasteiz, which are only distance 70 km but present clear meteorological differences due to its proximity to the coast (Bilbao is at about 15 km, and Vitoria at about 65 km) and the physical separation between them (Gorbea mountain chain).”

“This analysis also allows the possibility to investigate the behaviour of 222Rn concentrations under different meteorological scenarios. In this sense, and in the context of the global warming, heatwave is an increasingly important weather phenomenon due to the significant contribution to air quality, heat-related morbidity and mortality [15, 16]. The scientific effort to know the characteristics and influence of this phenomenon has been increased [17,18], i.e.,  climatic projections predict more intense, prolonged and frequent extreme heatwaves in Europe in the 21st century, with a higher impact on the Iberian Peninsula and Mediterranean regions [19]. To the best of the author's knowledge, no study has been conducted to investigate the behaviour of 222Rn concentrations under heatwaves. For this reason, the second objective of the present paper paper is to analysis the temporal variability of 222Rn concentrations under heatwave events registered in the Basque Country to evaluate potential health hazards due to radon inhalation under this extreme meteorological event.”

  1. Methodology does not describe much of the actual methodology

We agree with your comment. In order to make clear the methodology use in the present study, we have performed the following:

  • We have removed Figure 2 in the present version.
  • We have included this new paragraph in this section “Radon is measured in the BAI9100 by ABPD-Compensation and Measurement principle. The radiological station located in Bilbao and Vitoria is a combined Particulate and Iodine Monitoring system Type 9850-6 (Moving Filter Monitor BAI 9100 D, IOD-131 monitor BAI9103-2 and Gamma Dose Rate Detector LB6360, Berthold trademark positioned on the roof of a building at 10 m above ground level. Briefly, radon is estimated based on its progeny. Since equilibrium factor measurements were not available, it was considered simply the output of the instrument being aware that referring to 222Rn concentration it refers to 222Rn estimated through its progeny. The reader is referred to [20, 21] to obtain more information about how the radon time series are obtained. In the present study, radon measurements continuously registered at 10-min interval from June 2012 to June 2015 are used. The selection of this period is based on the large percentage of radon measurements taken at the same time in Bilbao and Vitoria. Hourly values are also calculated for the present analysis applying the sufficiency criterion (at least 75% of the 10-min and hourly values).”

  1. The description of the study areas is very cursory, there is no geology of the region, geomorphology,

Thanks for this comment. We have created a new section to describe the study area. In addition, we have included the following information ““According with MARNA, which is the Map of natural gamma radiation in Spain, both areas could be classified as low natural radioactivity background based on the value of U contents in soil, gamma dose rate and radon fluxes. Bilbao area and Vitoria area show similar uranium values between 1.6 and 2.4 mg/kg in the European Map of Uranium in soil (https://remon.jrc.ec.europa.eu/About/Atlas-of-Natural-Radiation/Digital-Atlas/Uranium-in-soil/Uranium-concentration-in-soil- ). So, those areas could be classified as low natural radioactivity background based on the value of U contents in soil, gamma dose rate and radon fluxes.”

  1. Applications must be supplemented with a comparison with the work of other researchers

We agree with your comment. We have included the following information in the new version of this manuscript:

  • “These averages agree with the typical outdoor radon concentration, between 1 and 100 Bq m-3, and with the estimated annual average of around 10 Bq m-3 [4].”
  • “Previous studies carried out have pointed out the link between the occurrence of sea-land breezes and the variability in 222Rn evolution and in the occurrence of high concentrations (Arrillaga et al., 2018; Gutiérrez-Álvarez et al., 2021). Specifically, in this region, Alegría et al., 2020, 2021 and Hernández-Ceballos et al., 2020 have revealed the positive link between this mesoscale phenomenon and high beta, alpha and 7Be activity concentrations”

  1. Literature is also too modest, no confrontation with other researchers

We understand your comment. We have performed the following modifications in the new version of this paper:

  • We have included in the introduction the following information “Previous studies have analysed radon outdoor concentrations measured in Bilbao [13], as well as indoor radon in Pozalagua cave [14,13]. However, no studies have yet addressed the spatial variability of radon measurements in this region. “
  • See previous comment
  • Unfortunately, this is the first paper in which the link between heatwave and radon variability is addressed, so there are not references about it to confrontate.
  •  

Other comments have been placed in the comments in the attached file.

We have copied your comment below:

Introduction

  • Cancel the date. A shorter title is more readable. In addition, the research is from a range of years

It has been done “Analysis of 222Rn surface concentrations in the Basque country (Spain): a case study of heatwave events”

  • Add keywords

It has been done “Keywords: radon; heatwave; northern Iberian Peninsula; air masses; local meteorology; climate change.”

  • What guided the authors when choosing research stations? eg: geological structure, increased content of radon, the presence of nuclear power plants?

This information has been included in the new version “For this reason, the first objective of this paper is to characterize radon concentrations in the Basque Country, taking as reference two monitoring sites, Bilbao and Vitoria-Gasteiz, which are only distance 70 km but present clear meteorological differences due to its proximity to the coast (Bilbao is at about 15 km, and Vitoria at about 65 km) and the physical separation between them (Gorbea mountain chain).”

  • Why is it so important to study this phenomenon, give a negative impact on people, construction.

We have included the following information “the second objective of the present paper paper is to analysis the temporal variability of 222Rn concentrations under heatwave events registered in the Basque Country to evaluate potential health hazards due to radon inhalation under this extreme meteorological event.”  

  • List the applicable standards to which the authors refer when analyzing their results

Thanks for this comment. For outdoor 222Rn values there are no such applicable standards because normal values are standardized. As example, WHO, in its page has the normal range between 1-10 Bq m-3, and another not normal range. (Radon and health (who.int). However, anomalous situations with atypical values are not normalized.

Materials and methods

  • The description of the regions should be a separate chapter, e.g. characteristics of the research area.

It has been done (see major comments)

  • In the materials and methods chapter, only refer to the methodology and possibly to the type of material tested

It has been done (see major comments)

  • Figure 1: Add the source of the presented mapDONE

It has been done “Figure 1. Location of the sampling site in the Basque country region (northern area of the Iberian Peninsula). Source: https://es-es.topographic-map.com/map-n3wtp/Península-ibérica/”

  • Add the device according to which the measurement was made, what were the technical parameters, standard

It has been done (see major comments)

  • What this figure represents, the description is unclear as well as the figure is unclear. Moreover, the descriptions in the figure are not in English and what is the point of posting this figure, if it is not exactly described in the methodology

We have removed this figure in the new version of the manuscript

Results

  • what about 2015

Sorry for this mistake. It has been changed

  • what for 1

It has been changed

  • Figure 4 - what the symbols mean, explain in the description

  • based on how many samples n=? DONE

This information is included at the beginning of the results “This method implies a selection of 660 “pair of days” during the period June 2012 to June 2015 to perform the comparison of 222Rn concentrations between Bilbao and Vitoria. The seasonal distribution of these days, 171 in winter, 143 in spring, 149 in summer and 197 in autumn guarantees the largest statistical sample.”

We have included the following information “Figure 5 shows the corresponding total and seasonal daily average 222Rn evolution for these two sites considering the 660 “pair of days” identified during the sampling period.”

  • Figure 5 - I understand that the black line is the average value, not the total

It has been changed to “Mean”

  • Where are the subsections 3.2, 3.3?

Sorry for the mistake. It has been changed

  • Figure 7: llegible, improve image quality, provide sources or map authors. What about Bilbao? After all, the analysis should cover both places.

This figure has been changed in order to improve the quality. We have only calculated backward trajectories in Vitoria due to its the city (province) in which heatwave events are officially registered. In addition, backward trajectories at 1500 m and 3000 m in Bilbao show the southerly displacement of trajectories. We have included the following sentence in the new version of the manuscript “Similar set of backward trajectories were obtained for Bilbao at these heights, which are the typical heights where Saharan air masses are observed over the Iberian Peninsula.”

  • so what are they related to? à We have included this comment to point out that heatwave events

We understand your comment. We are pointing out a result of the present study, i.e., in concentrations heatwaves are not related to the highest concentrations. We are then investigating the variability

Figure 8: The quality of the figure is very poor and illegible. Provide a more detailed description of these graphs in the title

We have changed it

  • Figure 9-10: note that the descriptions of these figures are illegible, just like the others

We have changed it

Conclusions

  • nothing new, no connection with the works of other authors

We have modified the conclusions. The confrontation with other works has been done in the results

“Radon concentrations collected between June 2012 and June 2015 in the Basque country region (North of Spain) have been studied. Similar averages were found in Bilbao (10.73 ± 0.08 Bq m-3) and Vitoria (10.1 ± 0.08  Bq m-3) although registering different seasonal patterns, i.e., in Bilbao the highest values were measured in summer and autumn, while in Vitoria, they were reached in winter and autumn.  These seasonal differences were observed in the analysis of daily cycles, which showed a similar trend in diurnal variations with differences in magnitudes. The role of wind direction in controlling the radon variability in time was also investigated. While in Bilbao, which is largely influence by maritime winds, the maximum concentrations are registered under flows from the south/southwest, in Vitoria, winds from the southwest/west are those responsible of the highest 222Rn activity concentrations. Finally, two specific time periods corresponding with heatwaves were used to investigate changes and variability of radon concentrations. Both events are not associated with the highest radon concentrations either Bilbao o Vitoria, but radon presents different temporal evolutions, and hence, it is not possible to identify a general pattern of radon concentrations under the occurrence of this synoptic scenario. The present analysis demonstrated the key influence of surface winds, and specifically, the development of sea-land breeze patterns in the coastal area on the temporal evolution of radon concentrations.”

Round 2

Reviewer 1 Report

Thank you for replying to my comments. The manuscript is ready to be published. 

Author Response

Comment 1: The following paragraph should be modified because of repeating sentence "areas could be classified as low natural radioactivity background based on the value of U contents in soil, gamma dose rate and radon fluxes". "According with MARNA, which is the Map of natural gamma radiation in Spain [24], both areas could be classified as low natural radioactivity background based on the value of U contents in soil, gamma dose rate and radon fluxes. Bilbao area and Vitoria area show similar uranium values between 1.6 and 2.4 mg/kg in the European Map of Uranium in soil (https://remon.jrc.ec.europa.eu/About/Atlas-of-Natural-Radiation/Digital-Atlas/Ura-nium-in-soil/Uranium-concentration-in-soil-). So, those areas could be classified as low natural radioactivity background based on the value of U contents in soil, gamma dose rate and radon fluxes.”

Many thanks for this comment, and we regret it. We have modified it in the new version of the manuscript “According with MARNA, which is the Map of natural gamma radiation in Spain [24], both areas could be classified as low natural radioactivity background based on the value of U contents in soil, gamma dose rate and radon fluxes. Bilbao area and Vitoria area show similar uranium values between 1.6 and 2.4 mg/kg in the European Map of Uranium in soil (https://remon.jrc.ec.europa.eu/About/Atlas-of-Natural-Radiation/Digital-Atlas/Uranium-in-soil/Uranium-concentration-in-soil-).”

Comment 2: "222Rn concentrations in Basque country (2012-2014)" but in the text is 2012 to 2015. Please correct.

Many thanks. It has been changed “4.1. 222Rn concentrations in Basque country (June 2012-June 2015)”

Comment 3: One of the goals of this paper was: "For this reason, the second objective of the present paper paper is to analysis the temporal variability of 222Rn concentrations under heatwave events registered in the Basque Country to evaluate potential health hazards due to radon inhalation under this extreme meterological event." But the authors did not present any results and conclusions regarding evaluation of health hazard. In addition, one word "paper" should be deleted.

You are right. We have modified it in the new version of the manuscript “For this reason, the second objective of the present paper is to analysis the temporal variability of 222Rn concentrations under heatwave events registered in the Basque Country to evaluate its behaviour under this extreme meteorological event.”
